# Predictors of Invasiveness in Adenocarcinoma of Lung with Lepidic Growth Pattern

**DOI:** 10.3390/medsci10030034

**Published:** 2022-06-22

**Authors:** Timothy J. Young, Ramin Salehi-Rad, Reza Ronaghi, Jane Yanagawa, Puja Shahrouki, Bianca E. Villegas, Brian Cone, Gregory A. Fishbein, William D. Wallace, Fereidoun Abtin, Igor Barjaktarevic

**Affiliations:** 1Division of Pulmonary and Critical Care, David Geffen School of Medicine, University of California, Los Angeles, CA 90095, USA; tjyoung@mednet.ucla.edu (T.J.Y.); rsalehirad@mednet.ucla.edu (R.S.-R.); rronaghi@mednet.ucla.edu (R.R.); 2Department of Surgery, David Geffen School of Medicine, University of California, Los Angeles, CA 90095, USA; jyanagawa@mednet.ucla.edu; 3Department of Radiology, David Geffen School of Medicine, University of California, Los Angeles, CA 90095, USA; pshahrouki@mednet.ucla.edu (P.S.); bevillegas@mednet.ucla.edu (B.E.V.); fabtin@mednet.ucla.edu (F.A.); 4Department of Pathology, David Geffen School of Medicine, University of California, Los Angeles, CA 90095, USA; bdcrq6@gmail.com (B.C.); gfishbein@mednet.ucla.edu (G.A.F.); 5Department of Pathology, Keck School of Medicine, University of Southern California, Los Angeles, CA 90033, USA; william.wallace@med.usc.edu

**Keywords:** lepidic pattern, lung cancer, ground-glass, adenocarcinoma in situ, lung biopsy

## Abstract

Lung adenocarcinoma with lepidic growth pattern (LPA) is characterized by tumor cell proliferation along intact alveolar walls, and further classified as adenocarcinoma in situ (AIS), minimally invasive adenocarcinoma (MIA) and invasive lepidic predominant adenocarcinoma (iLPA). Accurate diagnosis of lepidic lesions is critical for appropriate prognostication and management as five-year survival in patients with iLPA is lower than in those with AIS and MIA. We aimed to evaluate the accuracy of CT-guided core needle lung biopsy classifying LPA lesions and identify clinical and radiologic predictors of invasive disease in biopsied lesions. Thirty-four cases of adenocarcinoma with non-invasive lepidic growth pattern on core biopsy pathology that subsequently were resected between 2011 and 2018 were identified. Invasive LPA vs. non-invasive LPA (AIS or MIA) was defined based on explant pathology. Histopathology of core biopsy and resected tumor specimens was compared for concordance, and clinical, radiologic and pathologic variables were analyzed to assess for correlation with invasive disease. The majority of explanted tumors (70.6%) revealed invasive disease. Asian race (*p* = 0.03), history of extrathoracic malignancy (*p* = 0.02) and absence of smoking history (*p* = 0.03) were associated with invasive disease. CT-measured tumor size was not associated with invasiveness (*p* = 0.15). CT appearance of density (*p* = 0.61), shape (*p* = 0.78), and margin (*p* = 0.24) did not demonstrate a significant difference between the two subgroups. Invasiveness of tumors with lepidic growth patterns can be underestimated on transthoracic core needle biopsies. Asian race, absence of smoking, and history of extrathoracic malignancy were associated with invasive disease.

## 1. Introduction

Non-small cell lung cancer (NSCLC) accounts for the majority of primary lung malignancies, with adenocarcinoma being the most common subtype. The International Association for the Study of Lung Cancer/American Thoracic Society/European Respiratory Society (IASLC/ATS/ERS) classification criteria [1] for adenocarcinomas are based on the predominant histologic pattern of a malignant epithelial tumor with glandular differentiation. Lepidic pattern growth is characterized by tumor cell proliferation along intact alveolar walls. Lung adenocarcinoma with lepidic growth pattern is further classified as adenocarcinoma in situ (AIS), minimally invasive adenocarcinoma (MIA) and invasive lepidic predominant adenocarcinoma (iLPA). Compared to other histologic subtypes of lung adenocarcinoma (LUAD), lepidic predominant adenocarcinomas (LPA), specifically those classified as AIS and MIA, have been shown to have a favorable prognosis with a nearly 100% five-year survival rate. However, those with iLPA have a modestly lower survival rate, ranging from 85.7 to 100% [2]. The 8th International Association Study of Lung Cancer TNM classification [3] staging project for lung cancer classified patients with AIS into stage 0, while patients with MIA were classified into stage IA1. Nevertheless, the integration of IASLC/ATS/ERS and AJCC [4] classifications suggests that AIS and MIA may be collectively classified based on favorable outcomes into stage 0, while routinely classifying iLPA as stage IA [5].

The discrepancies in survival rates between AIS, MIA and iLPA are relevant considering that clinical management may be determined by the presumption of non-invasiveness or low-grade malignancy based on imaging [6]. While lobectomy is the standard for resectable LUAD, sublobar resection has been a proposed option in those with a lepidic predominant growth pattern, especially patients over 75 years of age and with tumors less than 2 cm [7]. Some authors have suggested that such lesions may not require mediastinal lymph node dissection [8], and hence, even favor non-surgical therapies such as ablation for these lesions [9]. However, the diagnosis of AIS or MIA cannot be confirmed without complete evaluation of tumor following surgical resection to exclude invasive disease [10,11]. Prior studies have reported concordance between core biopsy and final resection pathologies ranging between 58.6% and 77% in subtyping LUAD [12,13,14,15]. Identifying clinical and radiological predictors of invasiveness in combination with core needle biopsy in LPA may permit non-surgical therapies or sublobar resection to be more readily offered.

In this study, we aimed to evaluate the ability of transthoracic lung biopsy to accurately classify lepidic growth pattern lesions and predict the invasiveness of biopsied lesions. As a secondary objective, we sought to identify clinical predictors of invasive LPA in comparison to AIS or MIA in a cohort of patients with an initial biopsy diagnosis of non-invasive lepidic pattern growth lesions.

## 2. Materials and Methods

The University of California, Los Angeles pathology archive was queried for CT-guided core biopsies of lung nodules or masses that were characterized as “adenocarcinoma with lepidic growth pattern only, with no invasive component present in the biopsy material” between 2011 and 2018. Thereafter, the cohort was identified by including only cases with matching subsequent resection specimens. In total, 34 cases were identified. Histopathological reports of the core biopsies and subsequently resected tumor specimens were compared to determine concordance, and the explant histopathology was used as the definitive interpretation of invasive LPA versus non-invasive LPA (AIS or MIA). The clinical and pathological data were retrospectively obtained from the electronic medical records. Radiological parameters were obtained and analyzed from the biopsy CT scan by a single radiologist with more than 15 years of imaging experience (FA). Area of tumor in each biopsy was calculated by multiplying the estimated percentage of tumor on H&E sections by the aggregate area of biopsy fragments as stated in the gross descriptions. The imaging parameters included size: the maximum bi-dimensional measurement in the axial plane; density: solid, part solid, non-solid or cavitary; margin: smooth, lobulated, spiculated or ill-defined; shape: round, oval or irregular; and location: right upper lobe, right middle lobe, right lower lobe, left upper lobe or left lower lobe. All data were analyzed to assess for correlation with invasive disease.

Statistical analysis was performed using SAS 9.3 (SAS Institute, Inc., Cary, NC, USA). Fisher exact test was used for categorical variables. Wilcoxon rank sum test was used for continuous variables. Bland–Altman analysis was used to evaluate the agreement between radiologic and pathologic sizing of explanted lesions. *p*-values of less than 0.05 were considered statistically significant. Continuous data were presented as means with standard deviation, and categorical data were presented as percentage. The study was approved by the University of California, Los Angeles Institutional Review Board, protocol number 17-001536.

## 3. Results

Thirty-four patients were identified to have undergone CT-guided core biopsy of a lung nodule with histologic diagnosis of non-invasive lepidic growth pattern adenocarcinoma, followed by surgical resection. Biopsies were performed using a trucut 20-gauge biopsy gun in all cases, and a median of eight samples (range 2–12) were taken for each biopsy. Surgical excision of tumors was completed after the core biopsy was performed within a median of 35 days (IQR 23, 55). On computed tomography, eight (23.5%) were non-solid lesions, 19 (55.9%) were part-solid, six (17.7%) were solid, and one (2.9%) was cavitary nodules or masses. Of 34 core biopsies showing non-invasive pathology, 24 (70.6%) patients had a final diagnosis of invasive LPA on explant pathology, while only 10 (29.4%) were found to have a final diagnosis consistent with AIS or MIA. Four (16.7%) of the invasive LPA cases were high-grade.

Demographic characteristics including age, sex and race were compared between non-invasive and invasive LPA subgroups (Table 1). While there was no statistically significant difference in sex and age between those with non-invasive disease vs. iLPA, race, specifically Asian descent, was associated with invasive disease (*p* = 0.03). In our cohort, 8/8 (100%) of Asians were diagnosed with iLPA, in contrast to 13/22 (59%) of Caucasians or 0/1 of Black patients.

Smoking history was inversely associated with invasive disease (*p* = 0.03), with the majority of non-smokers (12/13, 92%) diagnosed with invasive disease in comparison to 57.1% (12/21) of smokers (Table 2). Active smoking and pack years were not associated with invasive disease. A history of unrelated extrathoracic cancer less than five years prior to biopsy of lung nodule was also associated with invasive disease (10/10 patients, *p* = 0.015). Presence of symptoms, such as cough or involuntary weight loss, were not associated with invasiveness (Table 2).

Radiographic features including density, margin, shape and location were not predictive of invasive disease (Table 3). However, there was a trend towards iLPA in solid tumors with irregular or round shape and ill-defined and spiculated margins. While the radiologic and explant size of the lesions correlated (R^2^ = 0.82, *p* < 0.0001) (Figure 1a), these measurements had poor agreement based on Bland–Altman analysis (*p* = 0.73) (Figure 1b). Although the median discrepancy between the radiologic and explanted size was 11 mm, with a wide range of over- and under-estimation of the actual size (IQR—5.0, 63 mm), there was no correlation between the time interval from biopsy to surgical resection and the extent of size discrepancy (R^2^ = 0.1, *p* = 0.6). The Mayo clinic model and Brock model for solitary pulmonary nodule malignancy risk were retrospectively applied and were not associated with invasiveness.

Histopathologic features including explant tumor size, presence of multiple foci, and presence of EGFR or KRAS mutations were not associated with invasive disease (Table 4). Thirty out of 34 cases had H&E sections from initial core biopsy available for tumor area calculation. The median area of tumor in the biopsies was 1.5 mm^2^ (range: 0.07 mm^2^–4.5 mm^2^). The mean tumor size of explanted nodules found to be iLPA was 26.1 ± 17.4 mm compared to 15.5 ± 6.7 mm in AIS and MIA lesions (*p* = 0.051), which demonstrated a non-statistically significant association between explant tumor size and invasive disease. Comparing radiologic vs. pathologic tumor size, 26.5% of cases were upstaged based on the pathology tumor size measurement (pTNM), with a majority of upstaged cases (66.6%) being invasive.

## 4. Discussion

In our cohort of patients with non-invasive LPA on CT-guided biopsy who subsequently underwent lung resection, the majority (70.6%) were diagnosed with invasive disease after tumor excision (Figure 2). The discordance between core needle biopsy and explant pathology demonstrated in this study reinforces the notion that the diagnosis of AIS or MIA cannot be confirmed without complete evaluation of the tumor [10,11].

Recent studies assessing the accuracy of core biopsy in lung adenocarcinoma subtyping based on final resection pathology have reported concordance ranging between 58.6% and 77% [12,13,14,15]. Our study is the first to report on discordance in core biopsy from final histopathology in differentiating non-invasive LPA from iLPA. While a favorable five-year survival rate is reported for the entire spectrum of tumors with lepidic predominant growth, there is a notable decrease in survival for those with invasive disease compared to AIS and MIA. Consequently, given the high proportion of invasive lesions found in our cohort, we confirmed previously published data [16] suggesting that the presence of non-invasive lepidic pattern should not be a reason for unnecessary delays in treatment for definitive diagnosis and management.

The small size of our cohort did not permit a more profound analysis of the predictors of invasiveness in this population of patients who underwent transthoracic lung biopsies. Nevertheless, our analysis of the relationship of clinical characteristics, radiological features and explant pathology results confirmed several previously reported findings. Asian race and non-smoking history have been reported to have a higher proportion of LPA over other subtypes of LUAD [16]. Our results show an association with invasive disease in these same subgroups independently. The majority of both smokers and non-smokers had invasive disease, 57.1% and 92%, respectively. However, the highest discrepancy between the core biopsy and tumor excision invasiveness was seen in never-smokers. These findings are consistent with what has been published recognizing the incidence of newly diagnosed lung cancer among non-smoking Asians [16,17]. We also found that history of extrathoracic malignancy within five years of diagnosis was associated with iLPA. We were unable to identify prior studies demonstrating an association of history of extrathoracic malignancy with iLPA, though it has been reported to predict a higher likelihood of lung cancer in a Chinese population with solitary pulmonary nodules [18]. Patients with a history of prior malignancy and NSCLC treated with surgical resection demonstrated worse five-year survival compared to those without a history of extrathoracic malignancy but without significant difference in lung cancer recurrence [19]. The clinical significance of our findings in regards to patients with LPA is unclear and warrants further evaluation in future studies.

The radiological features of LPA have been readily identified in the literature. Lepidic growth is characterized by ground glass nodules (GGNs) on CT imaging, and the degree of concurrent solid component can suggest invasiveness, with MIA lesions having equal or less than 5 mm and invasive LPA lesions having more than 5 mm of solid tissue on imaging [1,20]. We did not identify a significant trend in radiologic size of nodule correlating with invasiveness in our cohort, though measurements were taken of the entire nodule and not discriminated by solid versus ground glass component. There was relatively poor agreement between radiologic and explant size in our cohort. This is not an unexpected finding, although interestingly, radiology almost evenly underestimated and overestimated the size of these tumors, despite the fact that the interval between the biopsy and excision varied. The histopathologic size of tumors suggested an association with invasiveness, though this did not meet statistical significance in our cohort (*p* = 0.051). This would concur with prior published data demonstrating that larger tumor size correlates with invasive disease [16].

The definition of LPA leads one to presume that the imaging features mirror pathology. In one study by Ko et al., the proportion of solid volume showed a trend (*p* = 0.051) in distinguishing invasive LPA compared to AIS and MIA combined. Specifically, the percentage solid volume averaged 14.5% ± 6.6 (95% CI: 10.3%, 18.7%) for iLPA and 8.2% ± 6.5 (95% CI: 2.7%, 13.7%) for AIS and MIA in combination [21]. In another study, by Miao et al. [22], preoperative HRCT and histological subtypes after resection of 190 patients with stage 1A lung adenocarcinoma were analyzed. These subtypes included atypical adenomatous hyperplasia (AAH), AIS, MIA and invasive adenocarcinoma. GGO proportion (*p* < 0.001), margin (*p* < 0.001), border definition (*p* = 0.015), pleural retraction (*p* < 0.001) and enhancement (*p* < 0.001) had statistically significant differences at four histological levels, and there were no significant differences in bubble lucency, shape, air bronchogram, vessel convergence sign, pleural thickening, lymphadenopathy and EGFR mutation.

A few authors have also evaluated the invasive nature of pure and mixed GGNs on explant tissue and demonstrated the extent of invasive disease in pure GGNs. Oh et al. [23] showed malignancy rates of pure and mixed GGN to be 70% and 89.6%, respectively, and Nakata et al. [24] showed similar rates of 71.4% and 89.6%, respectively. However, in our study we did not see a statistically significant difference based on density, shape or margin between iLPA and non-invasive LPA (AIS and MIA). A possible explanation for this discrepancy is that the patients in our cohort were referred to biopsy, and a selection bias is possible where our patients may have had more worrisome clinical or radiological features that would prompt a biopsy.

Travis et al. [1] showed that biopsy samples may underrepresent the entire tumor biology, and a diagnosis of AIS or MIA cannot be firmly established without histologic sampling of the entire tumor. They concluded that to definitively assess LPA for true invasiveness, lesions must be surgically resected. Here, we draw the same conclusion, where 70.6% of biopsied samples were upstaged to iLPA after surgical resection. However, our study uniquely demonstrated the inability of imaging to predict the invasive nature of lesions, as there was no significant difference in imaging parameters between concordant and discordant cases on imaging (Table 3).

These findings confirm the recommendations of the IASLC/ATS/ERS that resection is an effective therapy for lesions with lepidic growth pattern [25]. While surgical management is a common approach, non-surgical options are often considered given an excellent prognosis [26]. While lobectomy has long been considered the gold standard in the treatment of lung cancer [27], there is controversy over whether lesions with a lepidic growth pattern can be treated with sublobar resection [28]. For its potential to impact surgical management, preoperative knowledge about the precise size and presence of the invasive component of a lepidic growth lesion may be of crucial relevance. Our data raise a concern that imaging characteristics accompanied by the core biopsy results may not be sufficient to provide precise information that would guide such surgical decisions.

There are several limitations to our study. This is a retrospective analysis of a small cohort of patients who underwent core needle biopsy prior to proceeding with surgical resection. As previously stated, there may be an inherent selection bias, where patients with more concerning clinical or radiographic features were referred for biopsy and patients who did not undergo biopsy prior to surgery were excluded, which may explain higher rates of invasive disease. Additionally, ground glass lesions are typically followed serially and noted to have changes in size prior to referral for biopsy and thus introduce a higher pre-test probability of invasive disease. Consequently, the prevalence of invasive lesions found here did not accurately reflect the actual prevalence of invasive lesions. This study does not offer a comprehensive assessment of clinical factors relevant for a complete estimation of pretest probability of lung malignancy, such as indications for biopsy or history and duration of previous follow-up of biopsied nodules.

## 5. Conclusions

Our data show that radiologic assessment combined with core biopsy analysis may underestimate the malignant potential of suspicious nodules with lepidic growth pattern. Asian race, non-smoking history and history of extrathoracic malignancy within five years are associated with invasiveness of these lesions. While the actual size of explanted lesions demonstrated a non-statistically significant trend towards the presence of invasive disease, the radiologic tumor size did not. Although one should consider the high pre-test probability of cancer directing these cases to biopsy, larger prospective studies and computer-aided texture analysis are necessary for further clarification of the nuances of the clinical approach to nodules diagnosed as non-invasive lepidic pattern growth tumors by needle biopsy.

## Figures and Tables

**Figure 1 medsci-10-00034-f001:**
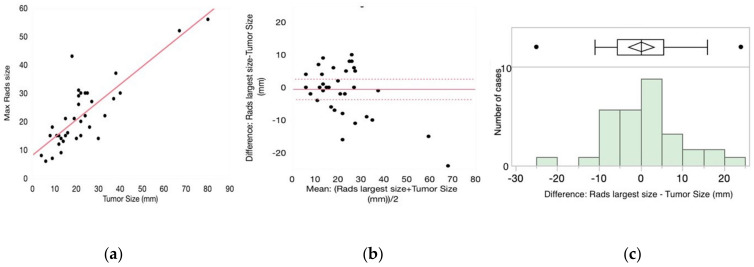
Comparison between the radiological and explanted total tumor size (invasive and noninvasive size). (**a**) Correlation between the radiological and explanted tumor size. (**b**) Bland-Altman analysis of the agreement between the two measurements. (**c**) Radiological size can both over- and under-estimate the actual explanted tumor size.

**Figure 2 medsci-10-00034-f002:**
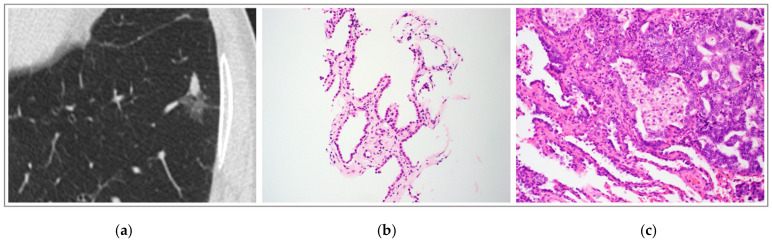
(**a**) Computed tomography of chest with part solid nodule in left lower lobe. Solid component of nodule had demonstrated interval growth. (**b**) Lung biopsy showing intact alveolar septa lined by atypical pneumocytes with hobnail appearance, hyperchromatic nuclei and scant cytoplasm. There is no interstitial, pleural or lymphatic invasion present (H&E, 200× magnification). (**c**) Lung resection showing areas of non-invasive lepidic growth (bottom left), immediately adjacent to areas of interstitial invasion (top right) (H&E, 200× magnification).

**Table 1 medsci-10-00034-t001:** Demographics.

	Invasive (*n* = 24)	Non-Invasive (*n* = 10)	*p*-Value	Total (*n* = 34)
Female, N (%)	14 (58.3)	6 (60.0)	0.62	20 (58.8)
Race, N (%)			** *0.03* **	
Asian	8 (33.3)	0 (0.0)		8 (23.5)
Black	0 (0.0)	1 (10.0)		1 (2.9)
Caucasian	13 (54.2)	9 (90.0)		22 (64.7)
Other	3 (12.5)	0 (0.0)		3 (8.8)
Age, Mean (SD)	70.3 (12.7)	69.2 (5.3)	0.30	69.9 (11.0)

**Table 2 medsci-10-00034-t002:** Clinical factors associated with invasiveness.

	Invasive (*n* = 24)	Non-Invasive (*n* = 10)	*p*-Value	Total (*n* = 34)
Smokers, N (%)	12 (50.0)	9 (90.0)	** *0.03* **	21 (61.8)
Pack-years, Mean (SD)	31.9 (0.2)	30.1 (0.3)	0.59	31.1 (0.2)
Current smoker, N (%)	2 (8.3)	0 (0.0)	0.31	2 (5.4)
FEV1%, Mean (SD)	94.8 (0.2)	103.7 (0.2)	0.2	97.3 (0.2)
DLCOHb%, Mean (SD)	79.9 (0.2)	75.7 (0.2)	0.08	78.7 (0.2)
Environmental exposures identified, N (%)	4 (16.7)	4 (40.0)	0.32	8 (23.5)
Family hx of lung ca, N (%)	7 (29.2)	3 (30.0)	0.67	10 (29.4)
Hx of extrathoracic ca less than 5 years prior, N (%)	10 (41.7)	0 (0.0)	** *0.02* **	10 (29.4)
History of COPD, N (%)	2 (8.3)	1 (10.0)	0.66	3 (8.8)
History of asthma, N (%)	3 (12.5)	0 (0.0)	0.34	3 (8.8)
Inhaler use, N (%)	3 (12.5)	1 (10.0)	0.66	4 (11.8)
Positive sputum culture in last 12 months, N (%)	1 (4.2)	0 (0.0)	0.71	1 (2.9)
History of cardiovascular disease, N (%)	5 (20.8)	4 (40.0)	0.23	9 (26.5)
History of CHF, N (%)	1 (4.2)	0 (0.0)	0.71	1 (2.9)
History of diabetes, N (%)	3 (12.5)	3 (30.0)	0.23	6 (17.7)
History of involuntary weight loss (5%/6 mos), N (%)	1 (4.2)	0 (0.0)	0.78	1 (2.9)
Daily cough, N (%)	5 (20.8)	1 (10.0)	0.42	6 (17.7)
Sputum, N (%)	0 (0.0)	0 (0.0)	n/a	0 (0.0)
Hemoptysis, N (%)	0 (0.0)	0 (0.0)	n/a	0 (0.0)
MTB Quantiferon positive, N (%)	1 (4.2)	0 (0.0)	0.67	1 (2.9)

Abbreviations: FEV1: forced expiratory volume in first second, DLCOHb: carbon monoxide diffusing capacity adjusted for hemoglobin, COPD: chronic obstructive pulmonary disease, CHF: congestive heart failure, MTB: Mycobacterium tuberculosis.

**Table 3 medsci-10-00034-t003:** Correlation of imaging with final pathology on explant lung.

	Invasive	Non-Invasive	*p*-Value
**Density, N (%)**			**0.61**
Cavitary	1 (4.2)	0 (0.0)	
Non-solid	4 (16.7)	4 (40.0)	
Part solid	14 (58.3)	5 (50.0)	
Solid	5 (20.8)	1 (10.0)	
**Margins, N (%)**			**0.24**
Ill defined	11 (45.8)	6 (60.0)	
Lobulated	4 (16.7)	0 (0.0)	
Smooth	1 (4.2)	2 (20.0)	
Spiculated	8 (33.3)	2 (20.0)	
**Shape, N (%)**			**0.78**
Irregular	12 (50.0)	6 (60.0)	
Oval	4 (16.7)	2 (20.0)	
Round	8 (33.3)	2 (20.0)	

**Table 4 medsci-10-00034-t004:** Pathology features assessed for association with invasiveness.

	Invasive (*n* = 24)	Non-Invasive (*n* = 10)	*p*-Value	Total (*n* = 34)
Tumor size (mm), Mean (SD)	26.1 (0.2)	15.5 (0.1)	0.051	23 (0.2)
Multiple foci, N (%)	3 (12.5)	2 (20.0)	0.47	5 (14.7)
EGFR mutation, N (%)	5 (20.8)	0 (0.0)	0.59	5 (14.7)
KRAS mutation, N (%)	5 (20.8)	1 (10.0)	0.55	6 (17.6)
Discrepancy rads vs path (>5mm), N (%)	15 (62.5)	5 (50.0)	0.38	20 (58.8)
Radiology overestimation of size (>5mm), N (%)	5 (20.8)	3 (30.0)	0.44	8 (23.5)

## Data Availability

The data presented in this study are included in the tables. Additional data are available on request from the corresponding author. The data are not publicly available due to patient privacy.

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
