# Peer review of "Predictors of Invasiveness in Adenocarcinoma of Lung with Lepidic Growth Pattern"

_medsci, 2022, doi:10.3390/medsci10030034_

Round 1

Reviewer 1 Report

Young et al. present data on 34 cases with lung core biopsies with only lepidic growth and subsequent resection from an 8-year period. To find factors predicting invasion in pre-surgical cases with lepidic growth is of clinical relevance. As a majority of cases exhibited invasion, it is clear that only lepidic growth in a biopsy does not rule out invasion.

The study includes a limited number of cases, which may affect the possibility to find factors predicting invasion, especially if low prevalence (as seen in Table 2).

Some further comments:

1) Total length of biopsies and total length of cancer in biopsies (in addition to number of samples per biopsied case) would be of interest. May it be that cases with many biopsies with extensive presence of lepidic AC are less commonly invasive?

2) Invasive lepidic predominant adenocarcinomas with at least 20% high-grade growth pattern are graded as high-grade according to IASLC/WHO 2021 edition. Were there any high-grade cases among the cases?

Reviewer 2 Report

Thank you very much for taking the time to consider my comments and suggestions.

Author Response

Thank you for your time in performing a thorough review of your paper and for your suggestions.

Round 2

Reviewer 1 Report

The manuscript has been amended as suggested. I have no further comments. 

This manuscript is a resubmission of an earlier submission. The following is a list of the peer review reports and author responses from that submission.

Round 1

Reviewer 1 Report

I would like to congratulate the authors for their interesting and informative paper.

This is a single-centre, retrospective study investigating the accuracy of CT-guided core needle biopsy of lung adenocarcinomas with lepidic growth pattern regarding their invasive component. The study included 34 patients, who were subsequently treated with surgical resection, and identified clinical and radiological predictors of invasiveness. The authors found that approximately 70% of explanted tumours demonstrated an invasive component. History of extrathoracic malignancy within 5 years, Asian race, and absence of smoking history were associated with invasive disease.

The overall quality of the manuscript is very good. The title describes the content of the paper adequately, and the introduction sets the appropriate background, even for the reader with little knowledge on the topic. The methods are sufficiently described to allow the conduction of replication studies, and the results are presented in a clinically meaningful manner. Finally, the conclusions answer the aim of the study and identify areas for future research.

Here, I would like to make some minor only suggestions.

  • Line 88: The authors may consider replacing the word “speculated” with the term “spiculated”.
  • Line 99: The authors may consider providing the approval number by the Institutional Review Board.

Finally, some misspellings and grammatical errors exist; these can be corrected at the proofreading process.

Reviewer 2 Report

Young et al. present data on 34 cases with lung core biopsies with only lepidic growth and subsequent resection from an 8-year period. To find factors predicting invasion in pre-surgical cases with lepidic growth is of clinical relevance. As a majority of cases exhibited invasion, it is clear that only lepidic growth in a biopsy does not rule out invasion.

The study includes a limited number of cases, which may affect the possibility to find factors predicting invasion, especially if low prevalence (as seen in Table 2).

Some further comments:

1) Total length of biopsies and total length of cancer in biopsies (in addition to number of samples per biopsied case) would be of interest. May it be that cases with many biopsies with extensive presence of lepidic AC are less commonly invasive?

2) Invasive lepidic predominant adenocarcinomas with at least 20% high-grade growth pattern are graded as high-grade according to IASLC/WHO 2021 edition. Were there any high-grade cases among the cases?

3) For table 2, abbreviations used in the table should be described under the table.